# Understanding the Role of Paper-Ink Interactions on the Lightfastness of Thermochromic Prints

**DOI:** 10.3390/ma16083225

**Published:** 2023-04-19

**Authors:** Rahela Kulčar, Marina Vukoje, Katarina Itrić Ivanda, Tomislav Cigula, Sonja Jamnicki Hanzer

**Affiliations:** University of Zagreb, Faculty of Graphic Arts, Getaldićeva 2, 10000 Zagreb, Croatia

**Keywords:** thermochromic printing, inks, light fastness, colour degradation, ink composition, spectroscopic studies

## Abstract

Thermochromic inks (TC) have received increasing attention in recent years, particularly in the design and packaging industries. Their stability and durability are crucial for their application. This study highlights the detrimental effects of UV radiation on the lightfastness and reversibility of thermochromic prints. Three commercially available TC inks with different activation temperatures and in different shades were printed on two different substrates, cellulose and polypropylene-based paper. Used inks were vegetable oil-based, mineral oil-based and UV-curable. The degradation of the TC prints was monitored using FTIR and fluorescence spectroscopy. Colorimetric properties were measured before and after exposure to UV radiation. The substrate with a phorus structure exhibited better colour stability, suggesting that the chemical composition and surface properties of the substrate play a crucial role in the overall stability of thermochromic prints. This can be explained by the ink penetration into the printing substrate. The penetration of the ink into the structure (cellulose fibres) protects the ink pigments from the negative effect of the UV radiation. Obtained results suggest that although the initial substrate may appear suitable for printing, its performance after ageing may not be optimal. In addition, the UV curable prints show better light stability than those made of mineral- and vegetable-based inks. In the field of printing technology, understanding the interplay between different printing substrates and inks is critical to achieve high-quality, long-lasting prints.

## 1. Introduction

During the last decades, thermochromic materials have received significant attention from both researchers and the industry [1]. Intelligent packaging today serves as a media for providing consumers with all the necessary information about the quality and safety of the packaging [2,3]. They are generating intense interest among artists and designers due to their interaction, responsiveness and ultimate functionality [4,5]. Thus, they offer unique and challenging design opportunities to the designers.

Thermochromic (TC) inks can convey a message to the consumer based on the colour of the ink they are seeing. Thermochromism is a phenomenon in which certain dyes made from organic or inorganic compounds and liquid crystals change colour when their temperature is changed [6]. The change may be reversible, i.e., the original colour may return upon cooling or irreversible. Similar colour changes, reversible or irreversible, are often observed when a thermochromic substance or system is cooled down. The mechanism responsible for the thermochromism of organic compounds varies with the molecular structure of the compound. The colour change is due to equilibrium changes, either between two molecular species (e.g., acid–base, keto–enol or lactim–lactam tautomerism), between different crystal structures, or between stereoisomers [7]. According to the studies, thermochromism occurs based on different mechanisms in which some of the following materials are used: inorganic compounds, organic compounds, polymers and sol-gels [8].

A thermochromic composite must be protected against undesirable influences from its surroundings, which is achieved by microencapsulation [9]. The polymer envelopes of the microcapsules are mostly made from epoxy or melamine resins, polyurea formaldehyde (PUF), polyphenolmelamine formaldehyde (PMF) or silicone [9,10,11]. Larger capsules are suitable for screen printing and smaller capsules for offset printing, where they have to face larger shearing forces during printing. Microcapsules are dispersed in a binder. The leuco dye-based thermochromic pigments consist of a colour former, a developer and a solvent in a specific combination. Common colour formers are electron-donating, leuco dyes, as exemplified by spirolactones, fluorans, spiropyrans or fulgides. The colour developer is a proton donor and normally is a weak acid such as bisphenol A, octyl p-hydroxybenzoate, methyl p-hydroxybenzoate, 1,2,3-triazoles, 4-hydroxycoumarin derivatives, laurylgallate, ethylgallate, p-hydroxybenzoic acid methyl ester and various other phenols, aromatic amines, carboxylic acids and Lewis acids. The melting point of the solvent controls the temperature at which the colour of the three-phase composite undergoes thermochromic changes. The solvent is a low-melting hydrophobic, long aliphatic chain fatty acid, amide or alcohol [12,13,14].

The leuco dye-based thermochromic pigments generally change from coloured to colourless or to another colour with an increase in temperature [15,16]. The colour change of leuco dyes occurs by two competing reactions, one between the dye and developer and the other between the solvent and developer. The first reaction prevails at low temperatures when the solvent is in solid form, so the dye-developer reaction prevails leading to the formation of dye-developer complexes [8]. In most cases, these complexes are coloured. When the solvent melts at a higher temperature, the solvent-developer interaction becomes dominant; thus, dye-developer complexes are destroyed, and the system converts into its colourless state. Additional components can be added to the thermochromic composite to obtain a wider range of colours [8].

The functionality of TC inks can be adversely affected by UV radiation, temperature above approximately 200–230 °C and aggressive solvents [17,18,19]. The colorimetric properties of those inks are only retained for a short period. The short pot life and poor stability of TC printing inks are related to the chemical stability of ink formulation. An acidic character of the binders, solvents and fountain solutions used during the printing process should be avoided. The main lack of thermochromic inks is their low light fastness. In the research by Pugh 2002, it was concluded that the number of pigment particles present determines the light stability of the print and not the actual particle size of the pigment [20]. Taking that in mind and the fact that TC microcapsules are 10 times larger than the conventional pigments, their amount in the formulation of printing inks is smaller in comparison to the number of conventional pigments. The UV absorbers were proposed as effective stabilisers preventing the photofading of leuco dyes and organic TC pigments [21,22,23]. Previous research has shown that the UV protective layer can help preserve the dynamic colour properties of TC prints [24,25] as well as the choice of printing substrate [26]. However, attention should also be given to the printing substrates used and to the chemical composition of the printing ink [26].

Thermochromic inks are very interesting and are increasingly used in various industries, including food packaging, promotional materials and security printing to create an eye-catching and interactive effect. In addition to their unique visual impact, their importance lies in their ability to convey information or indicate temperature changes. For example, a label printed with thermochromic ink on food packaging can indicate when the product is at the optimal temperature for consumption or when it has been exposed to excessive heat. In addition, the design possibilities for thermochromic inks are endless, as they can be customised to change colour at specific temperature ranges. By incorporating thermochromic inks into their designs, designers and brands can create unique and engaging experiences for their customers while communicating important information. However, because they are sensitive to UV radiation, there is a risk that the ink will degrade, resulting in a loss of print quality and functionality. It is therefore important to understand how the chemical composition of thermochromic inks and substrates affects their stability and print quality. Understanding these factors can be of great benefit to manufacturers and printers when developing new inks and selecting printing processes and substrates to produce high-quality, durable graphic products.

The novelty of the research is focused on understanding the interplay and interactions between thermochromic inks and printing substrates to achieve high-quality, durable products. The chemical composition of the inks and substrates are important factors that can affect the stability and print quality of the final product. By gaining a better understanding of these factors, manufacturers and printers can develop new inks and select printing processes and substrates that produce long-lasting, high-quality graphic products. This study specifically examines the interactions between TC inks and two different printing substrates.

## 2. Materials and Methods

### 2.1. Materials

Three commercially available TC inks with different activation temperatures (27, 31 and 45 °C) in different shades were tested. The producer’s data are presented in Table 1.

Two different printing substrates were printed with offset (TC45 and TC27) and screen printing thermochromic ink (TC31). Used inks were vegetable oil-based (TC45) and mineral oil-based (TC27), while the screen-printing ink was of the UV-curing type. Two printing substrates were used for printing, one made of polypropylene (PP) (Yupo, Tokyo, Japan, 73 g/m^2^) while another was cellulose-based paper (CP) (Munken Print White, Artic Paper, Munkedal, Sweden, 80 g/m^2^) (containing mechanical wood pulp). These two printing substrates were selected due to their different chemical composition, surface properties and ink absorption capacity. All the samples, before any analysis, were preconditioned at standard atmospheric conditions according to ISO 187:2022 [27].

### 2.2. Determination of Surface Roughness

All samples were analysed in terms of surface roughness (R_a_) using a surface roughness tester TR200 (Qualitest, Bridgewater, NJ, USA) according to ISO 21920-2:2021standard [28]. Average values of ten measurements on different places of the same sample were taken and presented as mean ± SD (Table 2).

### 2.3. Printing Trials

Two different techniques were used for printing samples: offset and screen printing. All samples were printed in full tone. For the offset printing trials, the Prüfbau Multipurpose printability Tester (Peissenberg, Germany) was used. A printing force of 600 N was applied while 1.5 cm^3^ ink was spread on the distribution rollers. Screen printing ink was applied using the Holzschuher K.G. (Wuppertal, Germany) screen-printing device, which used a 60/64 mesh and dried using the Technigraf Aktiprint L 10-1 device (Technigraf GmbH, Grävenwiesbach, Germany) under UV radiation of 30 W/cm.

### 2.4. Determination of Lightfastness

Lightfastness experiments were carried out in a Solarbox 1500e Xenon Test Chamber (CO.FO.ME.GRA, Milano, Italy) where the printed samples were exposed to 550 W/m^2^ UV radiation for a duration of 6 and 12 h. The test was conducted in accordance with standard procedures [29,30]. The time of UV radiation exposure of the samples was chosen regarding the stability of the print colour itself, which on some samples was significantly changed after 12 h.

### 2.5. FTIR Spectroscopy

The ATR spectra of the tested samples, unexposed and exposed after 6 and 12 h, were measured using Shimadzu FTIR IRAffinity-21 spectrometer (Shimadzu Corporation, Tokyo, Japan) with the Specac Silver Gate Evolution (Specac Ltd., Orpington, UK) as a single reflection ATR sampling accessory with a ZnSe flat crystal plate (index of refraction 2.4). The IR spectra were recorded in the spectral range between 4000 and 400 cm^−1^ at a 4 cm^−1^ resolution and averaged over 15 scans. The FTIR spectra are vertically displayed for visual clarity.

### 2.6. Colorimetric Spectroscopy

Colorimetric properties of printed samples before and after exposure to UV radiation were measured using an Ocean Optics USB2000+ spectrometer (Ocean Optics, Orlando, FL, USA) using a 50 mm wide integrating sphere (ISP-30-6-R) under (8:di) measuring geometry (diffuse geometry, specular component included). The printed samples were heated/cooled on the full-cover water block (EK Water Blocks, EKWB d.o.o., Komenda, Slovenia). Its temperature was varied by a thermostatically controlled water block. Reflectance spectra were measured in one heating/cooling cycle by heating them and then cooling them back to the initial temperature. The selection of heating/cooling cycles depended on the activation temperature of each colour being tested. Around T_A_, the reflectance spectra were measured in 1 °C intervals, but larger temperature differences were allowed (2 °C or 5 °C) elsewhere. A heating/cooling rate of about 0.5 °C/min was applied.

The measurements were performed in steps of 1 nm for the spectral region from 400 to 750 nm. Ocean Optics software (version 2.0.8) was used for the calculation of the CIELAB values from the measured reflectance. The D50 illuminant and 2° standard observer were applied in these calculations. Colour differences were calculated using the CIEDE2000 total colour difference formula according to Equation (1) [31].
(1)$${∆E}_{00}^{*}=\sqrt{{\left(\frac{{∆E}^{\prime }}{k_{L}S_{L}}\right)}^{2}+{\left(\frac{{∆C}^{\prime }}{k_{C}S_{C}}\right)}^{2}+{\left(\frac{{∆H}^{\prime }}{k_{H}S_{H}}\right)}^{2}+R_{T}\cdot \frac{{∆C}^{\prime }}{k_{C}S_{C}}\cdot \frac{{∆H}^{\prime }}{k_{H}S_{H}}}$$

### 2.7. Fluorescence Spectroscopy

To conduct fluorescence measurements, the combination of the spectrometer and integrating sphere (Ocean Optics, Orlando, FL, USA) used for UV-Vis spectroscopy were employed, along with an LSM Series (Ocean Insight, Orlando, FL, USA) LED light source at 365 nm. The LED light source was operated using a smart controller throughout the measurement, with a constant current of 0.140 A. Fluorescence intensity was measured within the spectral range of 330 to 630 nm.

## 3. Results

### 3.1. Surface Roughness Analysis

Table 2 shows the results of surface roughness determination. It can be seen that the cellulose-based paper has a heterogeneous surface with a rather high surface roughness compared to the polypropylene-based printing substrate. In addition, a polymer layer is likely to form when the ink is applied, resulting in lower surface roughness, especially for UV-curable inks (TC 31). For offset printing inks, it can be assumed that the ink is likely to penetrate the structure of the cellulose paper, but some of the ink will remain on the paper surface. This results in a rougher surface compared to UV-cured ink. Polypropylene-based substrates have a smooth surface that does not change significantly with ink application.

### 3.2. FTIR Spectroscopy of Thermochromic Ink and Papers

The light stability of thermochromic inks depends on the stability of the thermochromic binder, the microcapsule shell and the thermochromic composite inside the microcapsule itself as well as conventional pigments. As described earlier, due to its complex structure, the stability of TC prints exists on different molecular levels while oxidation takes place due to a variety of photochemical reactions [19]. To make an overall conclusion, and to explore the role of TC printing ink binder on UV stability, three different TC printing inks were printed on two different printing substrates. The degradation process was monitored with FTIR spectroscopy.

Figure 1 shows the FTIR spectra of cellulose-based and polypropylene-based papers. The IR spectrum of CP paper (Figure 1a) corresponds to the spectrum of cellulose, with common bands at 3310 cm^−1^ corresponding to OH stretching vibrations and antisymmetric and symmetric stretching of C-H bonds of methylene groups present at 2895 cm^−1^. The fingerprint region of CP paper between 1500 and 1300 cm^−1^ corresponds to deformation vibrations of the cellulose group, while bands between 1160 and 950 cm^−1^ correspond to the stretching of the C-C and C-O glucopyranose ring. The vibrational band assigned to C–O–C stretching at β-(1-4)-glycosidic linkages at 893 cm^−1^ is called an “amorphous” absorption band, while the vibrational band assigned to a symmetric CH_2_ bending vibration observed around 1429 ± 1 cm^−1^, is known as the “crystallinity band [32,33]. The presence of fillers in the paper, in this case the CaCO_3_, can be described with three bands at 1418, 874 and 709 cm^−1^ [34,35]. The presence of a lignin vibrational band at 1508 cm^−1^ (C=C of aromatic) points to the mechanical pulp used for the production of used paper [36]. Due to the complex structure of the paper (cellulose, hemicellulose, lignin, fillers and sizing agents), the vibrational bands very often overlap as in the case of the crystallinity band and asymmetric stretching CO_3_ vibration at around 1418 cm^−1^. With exposure to UV radiation (Figure 1a), the changes occurring in the paper point to changes in the cellulose structure by the rearrangement of hydrogen bonding (changes of vibrational bands in the range from 1300–1340 cm^−1^). The IR spectrum of the polypropylene-based printing substrate (Figure 1b) confirms its structure, namely by the presence of vibrational bands located at 839 cm^−1^(C–CH_3_ stretching vibration), the bands at 970, 997 and 1161 cm^−1^ (–CH_3_ rocking vibration), the band at 1375 cm^−1^ (symmetric bending vibration mode of –CH_3_), the bands located at 1455 cm^−1^ (–CH_2_– symmetric bending), 2838 cm^−1^ (–CH_2_– symmetric stretching) and 2917 cm^−1^ (–CH_2_– asymmetric stretching) [37]. With the exposure to UV radiation, no significant changes occur, pointing towards the conclusion that the exposure time used in this study does not affect its surface structure (Figure 1b).

FTIR analysis of prints shows the vibrational modes characteristic for the surface layer, i.e., a surface layer of ink in the binder (Figure 2, Figure 3 and Figure 4). Earlier studies showed that the screen-printing ink used, TC31, is based on polyurethane acrylate as confirmed in earlier studies [38,39] and by the presence of vibrational bands shown in Table 3. On the contrary, the used TC27 ink is oil-based [40] since its vibrational bands correspond to vibrational bands of vegetable and mineral oils (Table 4) [41,42,43]. The TC45 ink used is based on vegetable oils [41,44,45]. The presence of these components in the printing inks determines its drying mechanism. For example, in TC27 ink, the presence of mineral oils points to drying by penetration, while the presence of vegetable oils points to drying by oxypolymerization. The use polyurethane acrylate resin in TC31 ink requires UV radiation for ink to cure by photopolymerization.

In the case of all prints, the ink penetrates into the surface of cellulose-based paper but stays over the surface on polypropylene-based printing substrate since it is not absorbent, as confirmed by the surface roughness measurements as well (Table 2). This results in sharper IR vibrational bands of printing ink binders. Mineral and vegetable oils used in the formulation of printing inks are mobile and can penetrate into the cellulose-based printing substrate [46]. Thus, a significant contribution of the cellulose bands can be noticed in FTIR spectra of TC prints for TC27 and TC45 ink printed on cellulose-based printing substrate, especially in the fingerprint region from 1450–1200 cm^−1^ and 1160–1000 cm^−1^ range (Figure 2 and Figure 3).

**Table 3 materials-16-03225-t003:** Vibrational bands of UV curable TC print and their assignations [47,48,49,50,51,52,53].

Band (cm^−1^)	Assignation	Component
3358	NH stretching	polyurethane
2955–2855	symmetric and asymmetric CH_2_ stretching	
1724	C=O stretching	
1365	C–N stretching	
1101	C–O–C stretching	
1531	C–NH bending	
1617, 1462	ring stretching of the phenyl moiety	
1636	double bond (C=C)	acrylates
810, 987, 1408	deformation of CH_2_=CH–	
1060, 1190, 1294	C–O stretching	
1462	–CH_2_– bending	polyurethane and acrylate
1271	C–N and C–O stretching	

**Table 4 materials-16-03225-t004:** Vibrational bands of mineral- and vegetable-based TC prints (TC27 and TC45) and their assignations [41,54,55].

Band (cm^−1^)	Assignation
3014	=C-H stretching
2916–2845	symmetric and asymmetric CH_2_ stretching
1737–1722	C=O stretching
1636–1531	C=C stretching
1463	CH, CH_2_ stretching
1234–1100	C-O-C stretching of the ester functionalities
997	C–O–C stretching

TC-31 print on CP- and PP-printing substrates show vibrational bands of polyurethane acrylate (Figure 2 and Table 3). Due to a partial penetration of the ink into the structure of the CP printing substrate (Figure 2a), vibrational bands are somewhat sharper in the print on the PP-printing substrate (Figure 2b). Moreover, during UV radiation the changes occurring are higher in the case of the TC print on a PP-printing substrate (Figure 2b) than on a cellulose-based printing substrate (Figure 2a). In the case of the TC31 print, during photooxidation, first the double bonds in residual acrylates (i.e., their photolysis) react, resulting in their disappearance or decrease, namely the bands located at 1635, 1408, 1294, 1190 and 987 cm^−1^. The vibrational band at 810 cm^−1^ remains the same during photo-oxidation. The changes of the urethane bands (1534 and 3358 cm^−1^), 1230 and 1190 cm^−1^ as well as the C–H bands (2951 cm^−1^ and 2861 cm^−1^) can be noticed as well [47,48,49,50,51,52,53]. The width of a strong carbonyl band cantered at 1724 cm^−1^ increased with exposure to UV radiation, pointing to the formation of oxidation products, namely the carbonyl and carboxylic groups.

In the TC27 and TC45 prints (Figure 3 and Figure 4), vibrational bands from 2925 to 2850 cm^−1^ (-CH, -CH_2_, and -CH_3_ stretching bonding vibration of aliphatic chains) correspond to oils (i.e., chains of fatty acids) (Table 4). In addition, vegetable oils are characterised by additional bands at 3014 cm^−1^, 1463 cm^−1^, 1166 cm^−1^, 1101 cm^−1^, and 709 cm^−1^. The carbonyl stretching band of the triglycerides (1737–1722 cm^−1^) is commonly accompanied by the bands at 1232, 1155 cm^−1^, and points to the presence of an ester group. During exposure to UV radiation, the ester carbonyl functional group shows the spread in width and movement towards lower wavenumbers, pointing to the formation of secondary oxidation products, namely carboxylic acids.

In the case of the TC45 print on cellulose-based paper, it can be seen that the formation of oxidative products resulted in the formation of new vibrational bands at 1649 and 1512 cm^−1^ and the widening of the carbonyl vibrational band at 1722 cm^−1^.

### 3.3. Colorimetric Properties

The CIELAB values of all samples were calculated by applying measured reflectance spectra during the heating/cooling cycle. From the results (Figure 5 and Figure 6), the impact of UV radiation on all prints in a relatively short time is clearly visible. The high degradation effects are already noticeable on samples that were exposed to UV radiation for 6h. The functionality of thermochromic inks is reduced with longer exposure to UV radiation, mainly on polypropylene-based paper. With longer exposure to UV radiation, the colour of the polypropylene-based paper changes to pale shades of blue. In contrast, the colour on cellulose-based paper, with longer exposure to UV radiation, becomes pale yellow, which is a consequence of the presence of lignin in the substrate.

On the samples of TC31 and CTI45 printed on cellulose-based substrate, even after 12 h, the TC effect is still present, although the dynamics of colour changes are much smaller, as seen in Figure 6. An explanation of the resulting differences can be interpreted by different paper structures. On polypropylene-based paper samples, there is no absorption of the binder into the substrate. All colour components remain on the surface of the polypropylene-based paper; whereas in the cellulose-based paper, the microcapsules, along with the binder, penetrate the paper structure. In this case, the microcapsules that penetrated the paper structure remained protected from UV radiation within the paper structure itself.

Based on Figure 7, Figure 8 and Figure 9, the path of colour change, during the heating/cooling process and hysteresis as a result, can be clearly observed. Full symbols indicate the heating path and the open cooling path. The accompanying graphs show that the colour path during heating and cooling is not the same. The reason is that thermochromic materials have a memory, that is, the colour at a certain temperature will not be the same if we reach that temperature by heating or cooling. A distinction is made between samples printed on cellulose-based and polypropylene-based paper. With polypropylene-based paper, a larger colour shift towards lighter colours is evident, and it can be observed that the thermochromic effect in some samples disappears after 6 h of UV exposure (TC31 on polypropylene-based paper and TC27 on both tested substrates). For cellulose-based paper, the thermochromic effect is still present on the TC31 sample, which was exposed for 6 h to UV radiation, but even after 12 h to UV exposure, the thermochromic effect is still noticeable. The most stable sample in the experiment was CTI45. After 6 h of UV radiation exposure on both substrates the thermochromic effect is still present with a slight shift in the brightness *L**, while on cellulose-based paper the thermochromic effect is present even after 12 h of UV exposure.

The total colour contrast (TCC) of thermochromic inks refers to the complete change in colour that occurs when the ink is exposed to a specific temperature range. It is a critical parameter that determines the functionality and effectiveness of thermochromic prints.

Measuring the total colour contrast of thermochromic inks is important for ensuring the quality and performance of printed materials and products that use thermochromic inks. The area of colour hysteresis may be represented by the colour difference (CIEDE2000) between heated and cooled states of the same sample as a function of temperature. This is illustrated in Figure 10. The corresponding data are given in Table 5 (Tmax, CIEDE2000max, TCC). Samples appear differently during the two reversible TC reactions. Higher CIEDE2000max shows larger colour change occurring at TC reaction. The largest colour change was obtained on the CP-45 sample. With a longer exposure of the samples to UV radiation, the TCC decreases, which means that the TC samples lose their most important function—the clearly visible colour change.

### 3.4. Fluorescence Spectroscopy

The fluorescence intensity spectra of unprinted cellulose-based substrates exhibit two peaks in the blue region of the spectra (Figure 11), at 439 and 480 nm, due to the presence of optical brighteners in paper [24]. However, polypropylene-based substrates do not exhibit any emission spectra in the visible part of the spectrum and reflect a high rate of UV light at the excitation wavelength due to the absence of fluorescent molecules.

Cellulose-based prints exhibit a higher absorption of excitation light at 365 nm due to the presence of optical brighteners within the paper substrate. Furthermore, exposure of prints to UV light does not affect the absorption rate, which remains constant for the specified combination of cellulose substrate and thermochromic ink. Notably, TC31 cellulose-based prints exhibit the highest absorption rate, while TC27 shows the lowest.

The difference in absorption rates can be attributed to the composition and drying mechanisms of the inks. TC27 and TC45 inks penetrate the substrate while drying, resulting in greater exposure of the optical brighteners and their ability to absorb excitation light. In the case of TC31, a UV drying ink, it is designed to absorb the specified wavelength to enhance drying (Figure 11). Fluorescence intensity spectra of the polypropylene print with TC-31 ink show a low-intensity band at 520–650 nm with a maximum peak at 563 nm. Exposure of the samples to UV radiation enhances the fluorescence intensity. The curing of UV ink is initiated via photoinitiators that generate free radicals. These radicals then react with an acrylate group, which in turn continues to react with further acrylates until they are stopped by an undesired reaction or until the system runs out of accessible acrylates [56]. Typically, 25% of the acrylates remain unreacted when curing at room temperature. During exposure to UV radiation, further polymerization occurs, leading to a reduction in the amount of available acrylates, as confirmed by FTIR analysis (Figure 2). While fluorescence is often observed in molecules with rigid structures, those with lower quantum efficiency or lacking rigidity may experience enhanced internal conversion rates, increasing the likelihood of radiationless deactivation. Prolonged exposure to UV radiation causes an increase in the rigidity of the ink film, which is manifested by fluorescence. Similar ink behaviour can be noticed on a cellulose-based substrate with the addition of bands characteristic for optical brighteners. Due to the noticeable penetration of ink binder into the substrate, lower fluorescence intensity levels can be observed.

The fluorescence intensity spectra of the TC27 print on synthetic paper does not show any visible emission. To measure the fluorescence of mineral oils, lower excitation wavelengths (≤300 nm) must be used since their emission maxima are located in the range of 360–380 nm [57]. There is a significant increase in the absorption of the excitation light after the exposure of the printed samples to UV light. In contrast, the fluorescence intensity spectra of TC27 ink print on cellulose-based printing substrate exhibit the characteristic peaks of the substrate’s optical brighteners. However, these peak intensities decrease with prolonged exposure to UV light (Figure 11).

Depending on the type of oil, a broad peak ranging from 400 to 550 nm can be observed [58,59], except for extra virgin olive oil which exhibits fluorescence peaks between 500 and 720 nm [60,61]. When printing with TC45 ink on cellulose-based paper, the fluorescence intensity spectra show a maximum peak at 510 nm, consistent with previous studies [24]. However, exposure of the samples to UV radiation leads to a decrease in fluorescence intensity and a shift towards lower wavelengths, with peak positions corresponding to the optical brighteners within the paper substrate. On polypropylene-based paper, the fluorescence intensity spectra of the CTI45 print show a low-intensity broad peak from 475–550 nm, which may be attributed to oxidation products of vegetable oil [62], as supported by FTIR analysis (Figure 4).

## 4. Conclusions

This study highlights the negative impact of UV radiation on the lightfastness and reversibility of coloured samples of TC prints. The research also demonstrates that the range and dynamics of the TC inks decrease with ageing on both cellulose and polypropylene substrates. Although the degradation of colour was similar on both substrates, the cellulose-based substrate exhibited better colour stability. Additionally, the study found that while the initial print was better on the polypropylene substrate, greater damage to the TC microcapsules and colour degradation occurred after ageing. These findings highlight the importance of selecting appropriate printing substrates and careful management of ageing to maintain the stability and durability of TC prints in various applications. The relatively short exposure time to UV radiation significantly affects the colour of the print as a consequence of colourants and binder degradation. The exposure time to UV radiation does not influence the printing substrate, and thus, the formation of photooxidative products resulting from printing substrate degradation cannot be a cause for colour degradation. This was confirmed by FTIR spectroscopy as well. Degradation of the binder can influence the colour stability, and thus the binder with the greatest degradation potential exhibits a higher colour degradation rate, as in the case of TC-27 prints. Therefore, the lowest stability was obtained for the mineral-based printing ink and polypropylene printing substrate (non-absorbent). In addition, when the ink penetrates into the printing substrate structure, the ink is more protected from the influence of UV radiation and shows lower colour degradation rate. The fluorescence spectra confirmed the FTIR spectroscopy, i.e., the formation of oxidative products from printing ink binder degradation. Results confirm that better stability of the thermochromic prints can be achieved by proper choice of printing substrates and ink formulation.

## Figures and Tables

**Figure 1 materials-16-03225-f001:**
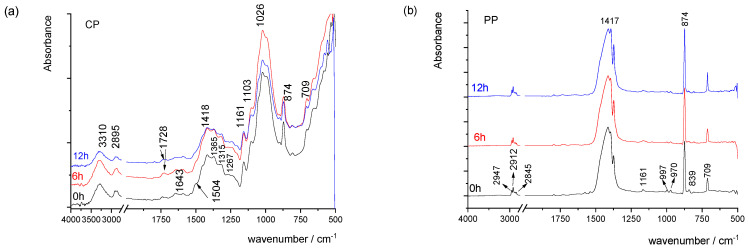
FTIR spectra of the cellulose-based (**a**) and polypropylene-based (**b**) paper before and after exposure to UV radiation.

**Figure 2 materials-16-03225-f002:**
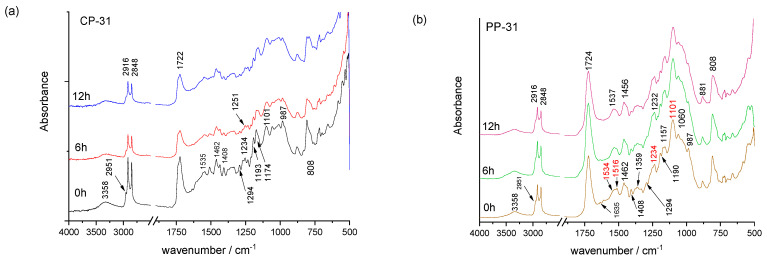
FTIR spectra of the TC31 ink on cellulose-based (**a**) and polypropylene-based (**b**) paper before and after exposure to UV radiation.

**Figure 3 materials-16-03225-f003:**
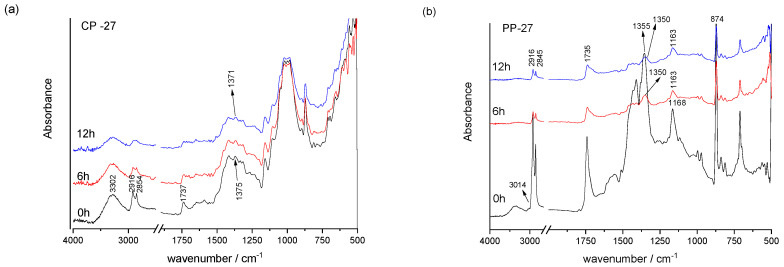
FTIR spectra of the TC27 ink on cellulose-based (**a**) and polypropylene-based (**b**) paper before and after exposure to UV radiation.

**Figure 4 materials-16-03225-f004:**
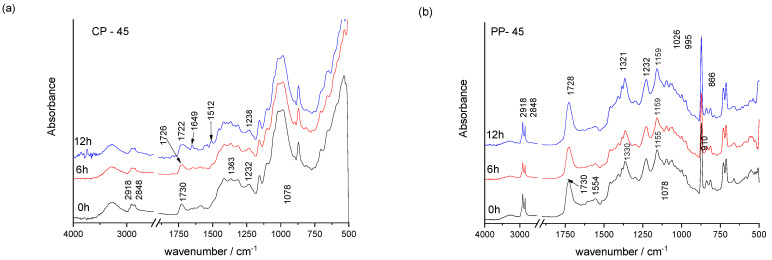
FTIR spectra of the TC45 ink on cellulose-based (**a**) and polypropylene-based (**b**) paper before and after exposure to UV radiation.

**Figure 5 materials-16-03225-f005:**
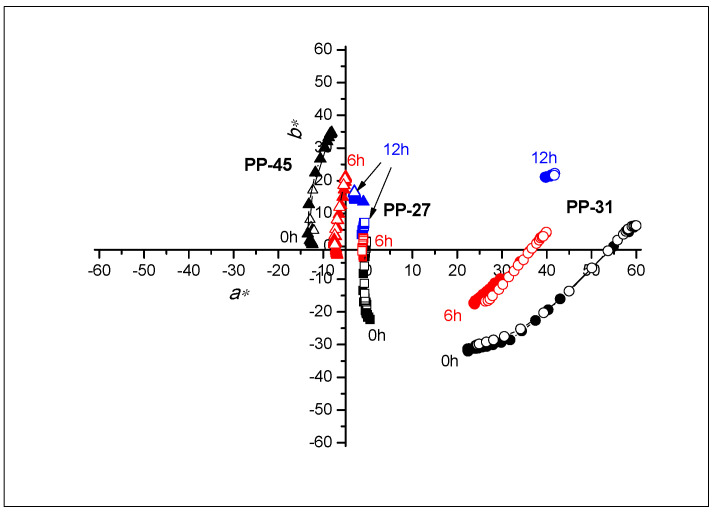
Changing of CIELAB values of all TC inks printed on polypropylene-based paper in the (*a**,*b**) plane at heating (solid signs) and cooling (open signs).

**Figure 6 materials-16-03225-f006:**
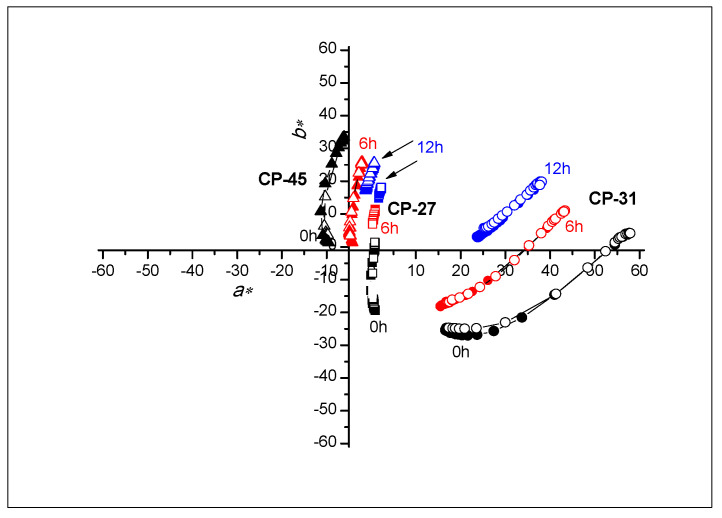
Changing of CIELAB values of all TC inks printed on cellulose-based paper in the (*a**,*b**) plane at heating (solid signs) and cooling (open signs).

**Figure 7 materials-16-03225-f007:**
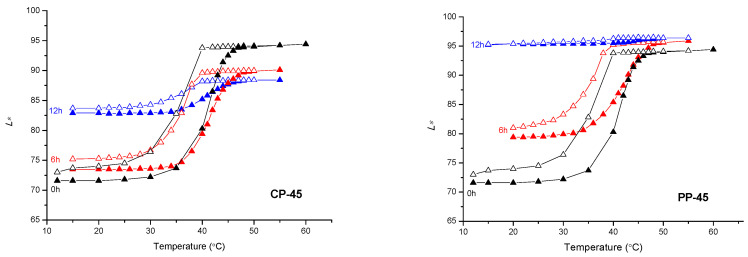
CIELAB lightness *L** of CTI45 sample in dependence on the temperature at heating (solid signs) and cooling (open signs) on both substrates.

**Figure 8 materials-16-03225-f008:**
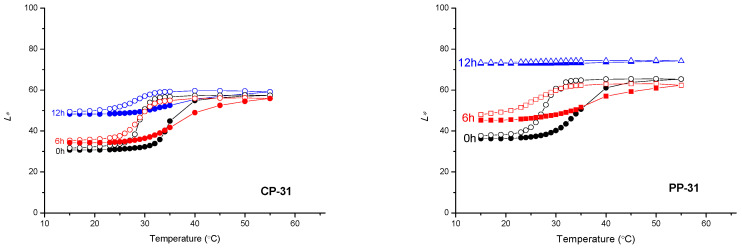
CIELAB lightness *L** of TC31 sample in dependence on the temperature at heating (solid signs) and cooling (open signs) on both substrates.

**Figure 9 materials-16-03225-f009:**
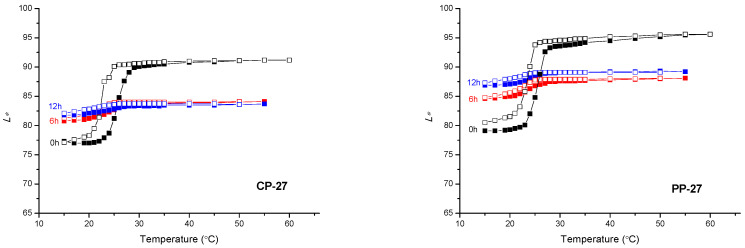
CIELAB lightness *L** of TC27 sample in dependence on the temperature at heating (solid signs) and cooling (open signs) on both printing substrates.

**Figure 10 materials-16-03225-f010:**
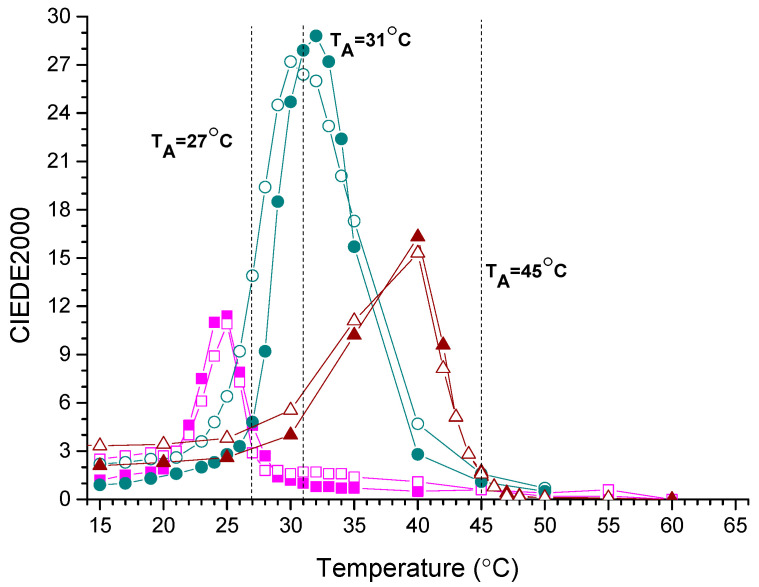
The total colour difference between the heated and cooled sample as measured for the TC27 (pink), TC31 (blue) and TC45 (red) samples (open signs represent PP printing substrate and solid CP printing substrate) before UV treatment, as a function of temperature.

**Figure 11 materials-16-03225-f011:**
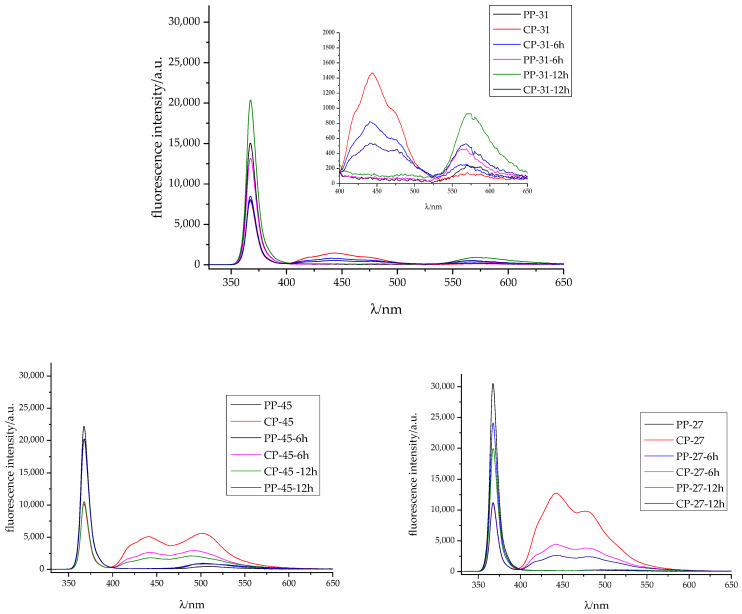
Fluorescence intensity spectra of the TC31, TC45 and TC27 prints on cellulose-based and polypropylene-based paper before and after exposure to UV radiation.

**Table 1 materials-16-03225-t001:** Properties of used thermochromic inks given by the producer.

Sample Abbreviation	Colour	TA, °C	Printing Technique	Drying Media	Drying Mechanism
TC45	green to yellow	45	offset	Air	Oxypolymerization
TC31	purple to pink	31	screen	UV	UV curing
TC27	blue to colourless	27	offset	Air	Penetration/Oxypolymerization

**Table 2 materials-16-03225-t002:** The roughness of unprinted and printed samples.

Printing Substrate	Roughness/µm			
	Unprinted	TC27	TC45	TC31
CP	4.01 ± 0.39	3.59 ± 0.31	3.74 ± 0.21	1.78 ± 0.24
PP	0.35 ± 0.05	0.33 ± 0.02	0.34 ± 0.03	0.23 ± 0.06

**Table 5 materials-16-03225-t005:** Tmax and CIEDE2000max describe the temperature dependence of the total colour difference between the heated and cooled sample and TCC total colour contrast between the sample at the lowest and the highest measured temperature at an original sample and after 6 and 12 h of UV radiation exposure.

Sample	Tmax (°C)	CIEDE2000max	TCC, Original	TCC, after 6 h	TCC, after 12 h
CP-31	32	28.8	34.8	29.6	14.6
PP-31	30	27.2	38.4	21.6	1.3
CP-45	40	16.3	24.3	19.7	19
PP-45	40	15.3	25.5	20.1	2.9
CP-27	25	11.4	17.6	4.5	2.4
PP-27	25	10.9	19.7	6.1	3.5

## Data Availability

Not applicable.

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
