# Peer review of "Understanding the Role of Paper-Ink Interactions on the Lightfastness of Thermochromic Prints"

_materials, 2023, doi:10.3390/ma16083225_

Round 1
Reviewer 1 Report
Dear Colleague
The following suggestions are provided to improve the manuscript:
1- Define the abbreviation in the first place i.e. Thermochromic inks in the abstract.
2- The engineering/scientific explanation provided in the results and discussion section is not satisfactory and hence to be revised thoroughly.
3- Very bad citing in introduction. Avoid bulk references.4- Novelty should emphasized in the introduction.
5- The ink company should mention.
6- Surface roughness method does not mention in the analysis.
7- Much more information regards all experimental methods is needed. There is lack of information on used apparatus, measuring techniques, sample preparations, etc.
8- With which peak the FTIR diagrams are normalized.
9- Abstract and conclusion need revision.
10- Please recheck all manuscript for grammatical, punctuations and other mistake.
11- Where is the paper-ink interactions effect? the manuscript is only the report of some commercial ink.
Author Response
We would like to thank the reviewers for pointing out the issues and changes required to improve the manuscript. We hope that we successfully responded the questions, improved suggested parts upon the Reviewers request. We hope that revised version of the manuscript meets the journal publication requirements.
All changes are made in Track changes.
Best regards,
The authors
Reviewer #1
Dear Colleague
The following suggestions are provided to improve the manuscript:
Comment 1: Define the abbreviation in the first place i.e. Thermochromic inks in the abstract.
Answer: The abbreviation was added in the abstract and in the Introduction part, lines 7 and 37.
Comment 2: The engineering/scientific explanation provided in the results and discussion section is not satisfactory and hence to be revised thoroughly.
Answer: We have carefully considered your comments on the Results and Discussion section of our paper, and we respectfully disagree with your assessment that the section needs to be thoroughly revised. We believe that our analysis and interpretation of the data are accurate, and we have provided a detailed and comprehensive discussion of our results. However, we recognize that scientific discourse is an ongoing process, and we are open to further constructive feedback. If there are specific areas of the section that you feel could be improved, please let us know and we will address them in our revision.
Comment 3: Very bad citing in introduction. Avoid bulk references.
Answer: Bulk references were corrected.
Comment 4: Novelty should emphasized in the introduction.
Answer: The novelty of the paper is written in lines 112-119.
Comment 5: The ink company should mention.
Answer: The used inks are made from Chameleon (TC27) and the other two are made from CTI (TC45 and TC31). Due to the GDPR, we are obligated not to reveal the company’s name, and due to that fact, we would like to keep the companies name private.
Comment 6: Surface roughness method does not mention in the analysis.
Answer: The method was mentioned at the end of the section 2.1. regarding the materials. Now it is moved to a new section 2.2. Determination of surface roughness – lines 137-141
Comment 7: Much more information regards all experimental methods is needed. There is lack of information on used apparatus, measuring techniques, sample preparations, etc.
Answer: Additional information have been added in the Experimental part.
Comment 8: With which peak the FTIR diagrams are normalized.
Answer: In this paper, qualitative analysis was done, not quantitative, therefore there was no need to normalize the FTIR spectra. Also, the research was conducted according to previous studies which prove to be good and reliable.
Normalization is a process used to correct for variations in intensity and is typically used in spectra where the intensity of the signal is directly proportional to the concentration of the analyte. However, in this study, the intensity of the signal is affected by many factors other than the concentration of the analyte, such as the thickness of the sample, the angle of incidence, and the refractive index of the sample. Normalization may be used in some cases to correct for variations in intensity, but it is not always necessary or appropriate for FTIR spectra. Instead, the focus in this study is on analyzing the shape and position of the peaks in the spectrum, which are indicative of the chemical functional groups present in the sample.
Comment 9: Abstract and conclusion need revision.
Answer: Abstract has been revised with the most important findings of the study.
Comment 10: Please recheck all manuscript for grammatical, punctuations and other mistake.
Answer: The manuscript has been improved in the terms of grammatical errors.
Comment 11: Where is the paper-ink interactions effect? the manuscript is only the report of some commercial ink.
Answer: The manuscript shows the combination of two different printing substrates – cellulose based (as a porous one) and polypropylene based (as non-porous one). For the porous paper, all inks penetrate into the structure while on polypropylene based, the ink remain on the printing substrate surface. This was also confirmed by the FTIR study and by the colorimetric measurements where it can be seen that the printing substrates affects the color characteristics. As a consequence of exposure to UV radiation, total color contrast was decreased on the non-porous printing substrate which does not have the protective property due to presence of TC microcapsules on the surface and higher exposure rate to UV radiation. As result of that TC effect degrades and decreases.
Reviewer 2 Report
The review report concerns about MS entitled “Understanding the role of paper-ink interactions on the light-2 fastness of thermochromic prints” is an excellent work. Overall, the MS is good written and within the scope of the journal “Materials”. Introduction section is well described. Results are well presented and discussed logically. Therefore, I recommend acceptation of this MS after the authors consider the following minor suggestions.
My suggestions are as follows:
1. Capital and small writing errors just recheck it throughout the manuscript, for example in line no 580 at reference no 62.
2. Please cite latest reference in introduction section.
3. Please check the space between word and recheck it throughout the manuscript for example in line no 472, 439.
4. Please check the Journal abbreviation reference no 15 line no 485.
5. Authors should check the grammatical errors throughout the manuscript,
Author Response
We would like to thank the reviewers for pointing out the issues and changes required to improve the manuscript. We hope that we successfully responded the questions, improved suggested parts upon the Reviewers request. We hope that revised version of the manuscript meets the journal publication requirements.
All changes are made in Track changes.
Best regards,
The authors
Reviewer #1
Dear Colleague
The following suggestions are provided to improve the manuscript:
Comment 1: Define the abbreviation in the first place i.e. Thermochromic inks in the abstract.
Answer: The abbreviation was added in the abstract and in the Introduction part, lines 7 and 37.
Comment 2: The engineering/scientific explanation provided in the results and discussion section is not satisfactory and hence to be revised thoroughly.
Answer: We have carefully considered your comments on the Results and Discussion section of our paper, and we respectfully disagree with your assessment that the section needs to be thoroughly revised. We believe that our analysis and interpretation of the data are accurate, and we have provided a detailed and comprehensive discussion of our results. However, we recognize that scientific discourse is an ongoing process, and we are open to further constructive feedback. If there are specific areas of the section that you feel could be improved, please let us know and we will address them in our revision.
Comment 3: Very bad citing in introduction. Avoid bulk references.
Answer: Bulk references were corrected.
Comment 4: Novelty should emphasized in the introduction.
Answer: The novelty of the paper is written in lines 112-119.
Comment 5: The ink company should mention.
Answer: The used inks are made from Chameleon (TC27) and the other two are made from CTI (TC45 and TC31). Due to the GDPR, we are obligated not to reveal the company’s name, and due to that fact, we would like to keep the companies name private.
Comment 6: Surface roughness method does not mention in the analysis.
Answer: The method was mentioned at the end of the section 2.1. regarding the materials. Now it is moved to a new section 2.2. Determination of surface roughness – lines 137-141
Comment 7: Much more information regards all experimental methods is needed. There is lack of information on used apparatus, measuring techniques, sample preparations, etc.
Answer: Additional information have been added in the Experimental part.
Comment 8: With which peak the FTIR diagrams are normalized.
Answer: In this paper, qualitative analysis was done, not quantitative, therefore there was no need to normalize the FTIR spectra. Also, the research was conducted according to previous studies which prove to be good and reliable.
Normalization is a process used to correct for variations in intensity and is typically used in spectra where the intensity of the signal is directly proportional to the concentration of the analyte. However, in this study, the intensity of the signal is affected by many factors other than the concentration of the analyte, such as the thickness of the sample, the angle of incidence, and the refractive index of the sample. Normalization may be used in some cases to correct for variations in intensity, but it is not always necessary or appropriate for FTIR spectra. Instead, the focus in this study is on analyzing the shape and position of the peaks in the spectrum, which are indicative of the chemical functional groups present in the sample.
Comment 9: Abstract and conclusion need revision.
Answer: Abstract has been revised with the most important findings of the study.
Comment 10: Please recheck all manuscript for grammatical, punctuations and other mistake.
Answer: The manuscript has been improved in the terms of grammatical errors.
Comment 11: Where is the paper-ink interactions effect? the manuscript is only the report of some commercial ink.
Answer: The manuscript shows the combination of two different printing substrates – cellulose based (as a porous one) and polypropylene based (as non-porous one). For the porous paper, all inks penetrate into the structure while on polypropylene based, the ink remain on the printing substrate surface. This was also confirmed by the FTIR study and by the colorimetric measurements where it can be seen that the printing substrates affects the color characteristics. As a consequence of exposure to UV radiation, total color contrast was decreased on the non-porous printing substrate which does not have the protective property due to presence of TC microcapsules on the surface and higher exposure rate to UV radiation. As result of that TC effect degrades and decreases.
Reviewer #2
The review report concerns about MS entitled “Understanding the role of paper-ink interactions on the light-2 fastness of thermochromic prints” is an excellent work. Overall, the MS is good written and within the scope of the journal “Materials”. Introduction section is well described. Results are well presented and discussed logically. Therefore, I recommend acceptation of this MS after the authors consider the following minor suggestions.
My suggestions are as follows:
Comment 1: Capital and small writing errors just recheck it throughout the manuscript, for example in line no 580 at reference no 62.
Answer: Correction has been made.
Comment 2: Please cite latest reference in introduction section.
Answer: Correction has been made.
Comment 3: Please check the space between word and recheck it throughout the manuscript for example in line no 472, 439.
Answer: Correction has been made.
Comment 4: Please check the Journal abbreviation reference no 15 line no 485.
Answer: Correction has been made.
Comment 5: Authors should check the grammatical errors throughout the manuscript,
Answer: Correction has been made.
Reviewer #3
The authors deal with the problem of stability and durability of thermochromic inks and the interaction between the ink and base. The article analyses the properties between the ink and the substrate The authors designed the experiment correctly, whereas the FTIR analyses were performed. The lightfastness of the prints was also determined. In my opinion, the paper can be accepted in its present form; however, prior the approval, the authors should consider some suggestions:
Comment 1: The authors should recalculate the exposition to UV radiation to the natural aging time;
Answer: The amount of UV exposure that a material receives is typically determined by measuring the combined effect of the intensity of UV radiation and the duration of the exposure. However, it's worth noting that the rate at which UV radiation damages a material is not constant over time. The material may experience an initial period of rapid degradation, followed by a slower period of damage accumulation as a result of changes to its chemical structure. For this reason, it is not appropriate to calculate UV exposure in terms of natural aging time, since this does not account for the non-linear nature of UV degradation or other contributing factors, such as exposure to moisture, heat, or other environmental stressors. While accelerated aging tests can help predict how materials will perform under real-world UV exposure, there is no single, reliable formula that can accurately predict the aging process from artificial to natural conditions. Nonetheless, it is important to note that while these tests can provide useful insights, they may not always offer an exact prediction of how a material will behave over time, and additional validation and testing may be necessary to ensure accuracy.
Answer:
Comment 2: Together with the samples, the Blue Wool Scale should be also placed in the test chamber;
Answer: The Blue Wool Scale and the ASTM D4303 standard test method can both be used for the evaluation of the colourfastness and lightfastness of materials.
We agree with the Review that Blue Wool Scale can be used as a indicator of degradation, and it can be the subject of future work. But our experiment was carried out in accordance with ASTM D 3424 - Designation: D 3424 – 01 Standard Test Methods for Evaluating the Relative Lightfastness and Weatherability of Printed Matter.
The Blue Wool Scale does not provide a quantitative measurement of colour change, nor does it simulate real-world exposure conditions. On the other hand, ASTM D 3424-01 is a standard test method specifically designed to evaluate the relative lightfastness and weatherability of printed matter. It suggests the use of xenon arc lamp to simulate exposure to natural daylight, which more closely mimics real-world exposure conditions. It measures the colour change of the colorant over time using a spectrophotometer and reports the results as a Delta E value, which provides a quantitative measurement of colour change. While the Blue Wool Scale may be used in some industries and applications to evaluate colourfastness, the ASTM D 3424-01 standard test method is a more widely accepted and reliable method for evaluating the lightfastness of prints. Thus, our study was conducted according to the ASTM D 3424-01 standard procedure.
Comment 3: The authors should highlight the novelty of the paper.
Answer: The novelty of the paper is written in lines 112-119.
Reviewer 3 Report
The authors deal with the problem of stability and durability of thermochromic inks and the interaction between the ink and base. The article analyzes the properties between the ink and the substrate The authors designed the experiment correctly, whereas the FTIR analyses were performed. The lightfastness of the prints was also determined. In my opinion, the paper can be accepted in its present form; however, prior the approval, the authors should consider some suggestions:
The authors should recalculate the exposition to UV radiation to the natural aging time;
Together with the samples, the Blue Wool Scale should be also placed in the test chamber;
The authors should highlight the novelty of the paper.
Author Response

(The authors gave the same response as above.)
